# Optimal Selection of IFN-α-Inducible Genes to Determine Type I Interferon Signature Improves the Diagnosis of Systemic Lupus Erythematosus

**DOI:** 10.3390/biomedicines11030864

**Published:** 2023-03-12

**Authors:** Veronique Demers-Mathieu

**Affiliations:** Independent Researcher, Escondido, CA 92029, USA; demersmv@gmail.com; Tel.: +1-541-286-8366

**Keywords:** IFNα, autoimmune disease, IFN-α-inducible genes, diagnostics, monitoring disease activity, autoantibodies, IFN-I, immunopathogenesis, inflammation, gene expression

## Abstract

Systemic lupus erythematosus (SLE) is a chronic autoimmune disease characterized by the production of autoantibodies specific to self-molecules in the nucleus, cytoplasm, and cell surface. The diversity of serologic and clinical manifestations observed in SLE patients challenges the development of diagnostics and tools for monitoring disease activity. Elevated type I interferon signature (IFN- I) in SLE leads to dysregulation of innate and adaptive immune function, resulting in autoantibodies production. The most common method to determine IFN-I signature is measuring the gene expression of several IFN-α-inducible genes (IFIGs) in blood samples and calculating a score. Optimal selection of IFIGs improves the sensitivity, specificity, and accuracy of the diagnosis of SLE. We describe the mechanisms of the immunopathogenesis of IFN-I signature (IFNα production) and its clinical consequences in SLE. In addition, we explore the association between IFN-I signature, the presence of autoantibodies, disease activity, medical therapy, and ethnicity. We discuss the presence of IFN-I signature in some patients with other autoimmune diseases, including rheumatoid arthritis, systemic and multiple sclerosis, Sjogren’s syndrome, and dermatomyositis. Prospective studies are required to assess the role of IFIG and the best combination of IFIGs to monitor SLE disease activity and drug treatments.

## 1. Introduction

Systemic lupus erythematosus (SLE) is an autoimmune disease that is characterized by the production of autoantibodies reactive to intracellular molecules, including nucleic acid and nucleic-acid-binding proteins, resulting in inflammation of many different organ systems, including skin, joints, kidney, blood, nervous system, and blood components [1]. The diversity of serologic and clinical symptoms observed in SLE patients (primarily women) challenges the development of diagnostics and new tools for monitoring disease activity [2]. New immunological markers are needed to improve prognostics and monitor SLE disease.

The most distinguishing feature of SLE is an immune response to nucleic acid and associated proteins, resulting in autoantibody production, immune complex formation, and organ inflammation. Intense complement activation and elevated type I interferon (IFN-I) (often referred to as IFNα) production are critical factors in the pathogenesis of SLE [3]. IFNα cytokines are major effectors in the pathogenesis of SLE, with several regulatory effects on both innate and adaptive immunity, resulting in the initiation and progression of SLE disease. Specifically, most SLE patients have elevated expression of IFN-α-inducible genes (IFIGs) compared to healthy controls and patients with rheumatoid arthritis (RA) [4]. Thus, elevated IFN-I expression contributes to the pathogenesis of SLE.

Elevated IFN-I signature in SLE induces dysregulation of innate and adaptive immune function (Figure 1). The clinical consequences are increased disease severity, hyperinflammation, and skin rash due to the high induction of apoptosis, chemokine production, and myeloid cell recruitment [5]. High IFN-I expression leads to immune-complex-mediated multiorgan damage and elevated autoantibody production due to abnormalities in B-cell differentiation. Patients with elevated IFN-I signature are more likely to develop lupus nephritis (the kidney is inflamed, with impaired function, leading to kidney failure). Moreover, the IFN-I signature promotes CD8 T-cell exhaustion and CD4 proliferation, leading to cellular infiltration, autoimmunity, and organ damage. Indeed, CD8^+^ T cells from peripheral blood of SLE frequently display a reduction in effector function, including attenuated granzyme B and perforin production. In addition to this impaired cytolytic defect in CD8^+^ cells, CD4^+^ follicular helper (Tfh) cells produce IL-21 and promote the germinal center formation of B cells and the differentiation of Th17 cells, which contribute to the pathogenesis of SLE.

## 2. Interferon-α-Inducible Genes (IFIGs) Used to Calculate Type I Interferon Signature

The four IFIGs frequently used to determine the IFN-I signature are IFI27, IFI44, IFI44L, and RSAD2 (Table 1) [4,6]. Other IFIGs used less frequently in IFN signature are IFIT1, LY6E, MX1, HERC5, EPSTI1, OAS3, OAS1, ISG15, PRKR, SIGLEC1, MX1, and HERC5. Leukocyte subsets that are related to IFN-I signature are CD19^+^ B lymphocytes, CD3^+^CD4^+^ T lymphocytes, and CD33^+^ myeloid cells [7]. IFIGs are expressed mainly in B cells, T cells, or/and myeloid cells (which include monocytes), and these immune cells are present in peripheral blood samples from SLE patients [7].

Some IFIGs are more expressed in patients with SLE than in patients with other autoimmune diseases, including RA. IFI44 and PRKR mRNA levels in peripheral blood mononuclear cells were higher in SLE patients than in RA patients and healthy donor controls using qPCR analysis, whereas IFIT1 was comparable between SLE and RA patients [4]. However, IFIT1 expression was higher in SLE patients compared to healthy controls (outside the setting of active viral infection) using RT-qPCR [4,8].

The most popular molecular methods available for the determination of IFN-I signature in SLE are RT-qPCR, microarrays, and Nanostring technology. Nanostring is useful for screening IFIGs, as it can generate expression analysis of hundreds of genes across many samples. Nanostring nCounter is a strongly sensitive technique, as it can detect low-abundance mRNAs that are not detected by DNA microarrays [9,10]. The sensitivity of the Nanostring nCounter is comparable to RT-qPCR (Taqman assays or SYBR Green) [9]. Results generated by Nanostring showed higher concordance with RT-qPCR than DNA microarray assay [9]. Gene expression analysis using microarrays is less precise than RT-qPCR, but there is a high degree of correlation between the two methods [11,12]. Lastly, Nanostring is a digital barcode technology for the direct multiplexed measurement of gene expression and is less time-consuming but more expensive than RT-qPCR [13].

When using RT-qPCR, the relative gene expression (RE) is typically determined using the 2^−∆∆CT^ method (C_T_, cycle threshold). Individual gene expression (∆C_T_) of each IFIG is normalized with one or more housekeeping genes (∆C_T IFIG_ = C_T IFIG_ − C_T Housekeeping_). The exponential expression (∆CqE) is calculated using 2^−1 × (∆CT IFIG)^. The relative gene expression (∆∆C_T_) of each IFIG of its corresponding sample is then normalized with the ∆C_T_ value measured in a pool of healthy control (HC) (RE ∆∆C_T IFIG-SLE_ = ∆C_T IFIG-SLE_ ÷ ∆C_T IFIG-HC_). The sum or median of the relative gene expression for all IFIGs is used to calculate the IFN-I score for each sample. Log_2_ can be applied to the final IFN-I score to obtain a value between 0 and 10.

A few studies have evaluated the sensitivity and specificity of the IFN-I signature in SLE patients and other autoimmune diseases. The IFN6 score (calculated with SIGLEC1, IFI27, IFI44L, IFIT1, ISG15, and RSAD2 relative expression) using RT-qPCR in SLE patients had 89% sensitivity and 72% specificity using pediatric patients in the interferonopathy group (SLE, dermatomyositis, and connective tissue disease) and non-interferonopathy (juvenile idiopathic arthritis and healthy controls) [13]. As seen in RT-qPCR, the IFN6 score using Nanostring resulted in a sensitivity of 90.5% and specificity of 63.3% [13]. On the other hand, the IFN5 score (calculated with EPST11, IFI44L, LY6E, OAS3, and RSAD2) using RT-qPCR had between 84% sensitivity and 47% specificity in 137 SLE patients [14].

For Nanostring nCounter, the IFN score can be calculated by two methods: a z-score-based standardized score and a geomean score. Standardized z-scores for each IFIG are determined using the mean and standard deviation of healthy controls (HC) with the following equation: (z − score for each gene = [(gene count − mean (HC gene expression)]/[standard deviation (HC gene expression)]. For example, the IFN28 standardized score was calculated by summing the 28 z-scores for each sample [10]. Alternatively, the geometric mean or geomean of the counts of each IFIG in each sample is calculated to generate the IFN score. In addition, the IFN5 score (EPSTI1, IFI44L, LY6E, OAS3, and RSAD2) was determined by Nanostring custom array with different ranges of sensitivity (63.6–83.8%) and specificity (41.7–67.2%) depending on the outcomes (disease activity state, intake of prednisone, ≥2 flares, and new immunosuppressive) [14].

For microarrays, the mean and standard deviation for three IFIGs (IFI27, OAS3, and IFI44) from 27 HCs were used to calculate the IFN3 score [11]. For each SLE patient, a z-score was calculated for these three IFIGs by subtracting the HC mean from the SLE expression value for that IFIG and then dividing the difference by the HC standard deviation. No sensitivity or specificity was reported in this study [11]. This scoring system has also been used in other studies on IFN-I signature using RT-qPCR [15,16].

More studies are needed to understand the importance of selected IFIGs to optimize and improve the sensitivity and specificity (>80%) for the diagnosis of IFN-I signature in SLE patients. Possible ways to improve these parameters are to increase the number of IFIGs (>6 IFIGs used to calculate IFN-I score) that are highly expressed in SLE patients compared to HC and other autoimmune diseases, such as rheumatoid arthritis (RA), systemic sclerosis (SSc), Sjögren’s syndrome (SS), dermatomyositis (DM), and multiple sclerosis (MS).

## 3. Mechanism of Type I Interferon Signature in the Pathogenesis of SLE

The first step of IFN-I signature in the pathogenesis of SLE is when plasmacytoid dendritic cells (pDCs) produce IFNα after immune recognition of autoantigens derived from apoptotic material from dead and dying cells and neutrophil extracellular trap debris (Figure 2) [17]. The production of IFNα promotes the maturation of monocytes to dendritic cells, activation of T cells, and simulation of B cells [18]. Thus, IFNα can play a role in the activation and differentiation of B cell into autoantibody-producing plasma cells and promote SLE disease. Although IFN-α is produced by a wide range of cells, such as macrophages, fibroblasts, and endothelial cells, plasmacytoid dendritic cells (pDCs) are thought to be the primary cell type responsible for producing high levels of IFN-α in response to dying cells.

A typical feature of SLE is lymphopenia and leukopenia, where T cell and B cell populations are reduced [1]. Endogenous production of IFNα has been implicated in the pathogenesis of leukopenia in SLE disease, as elevated serum levels of IFNα in SLE were correlated negatively with leukocyte numbers [18,19]. Absolute lymphocyte count was 1.3-fold lower in high IFNα scores than in low IFNα scores in SLE patients, whereas the percentage of monocytes was 1.6-fold higher in the high IFNα score group [4]. Leukocyte counts were also correlated negatively with IFNα concentration in serum [20]. In addition, the transcriptional activity of leukocytes (CD19^+^CD3^−^ B cells and CD3^+^CD4^+^ T cells) was higher in SLE patients than in healthy controls [7], suggesting the prolonged upregulation of nucleic-acid-sensing pathways could change the immune effector functions and induce a systemic inflammation observed in SLE pathogenesis.

## 4. Higher Production of Autoantibodies in SLE with Elevated IFN-I Signature

In SLE, the immune component complexed with RNA or DNA promotes plasmacytoid dendritic cells to produce IFNα by activating toll-like receptors (TLR) TLR7 or TLR9, respectively. IFNα score correlates positively with autoantibodies in SLE patients, including anti-double-stranded DNA (dsDNA) and anti-Smith (Sm) antibodies [21,22]. Moreover, IFNα enhances the differentiation of T follicular helper (Tfh) and Th1 cells, activating the production of autoantibodies by plasma cells and elevating tissue inflammation [23].

Antinuclear antibodies (ANA) consist of various autoantibodies targeting nuclear protein and cytoplasmic cell components, including anti-dsDNA, anti-ribonucleoprotein (RNP), anti-Sm, and anti-Ribosomal P (Rib-P) antibodies, and are detected in ~95% of SLE patients (Table 2) [24]. ANA are not specific to SLE, as they can be detected in different autoimmune, rheumatic, and infectious diseases [25]. Anti-dsDNA antibody is among the most specific autoantibodies in patients with SLE but is not sensitive, as its frequency is only 50–60%. Similarly, anti-Sm and anti-RNP are less frequently detected in SLE than anti-dsDNA antibodies in SLE disease, while anti-Rib-P antibodies are even less sensitive than those autoantibodies. Anti-Sm antibodies are specific to core proteins, especially B protein followed by D1 and D2 [26]. Anti-RNP antibodies are directed against A and C proteins that are associated with the U1 RNA (U1-RNP complex) [26]. Interestingly, anti-dsDNA IgG, anti-RNP IgG, and anti-C1q antibodies were associated with disease activity in SLE patients [27].

SLE patients with anti-RNP and anti-Sm antibodies were more prevalent in the IFN3α high-score group than in the low-score group in SLE (calculated with IFI44, IFIT1, and PRKR). In contrast, the levels of anti-RNA-binding protein (RBP) and anti-dsDNA antibodies were comparable between the two groups [4]. Anti-RBP antibodies can bind to different RNA-binding proteins, including Sm and RNP. Similarly, the IFN3 score (calculated with LY6E, OAS1, and IFIT1) was positively correlated with the levels of anti-dsDNA and anti-RBP antibodies in blood from SLE patients [21]. A high IFN3 score (calculated with MX1, PKR, and IFIT1) was associated with the presence of anti-dsDNA and anti-RBP antibodies in SLE [22]. In addition, the ANA titer was correlated positively with an IFN63 score (averaging 63 IFIGs) in SLE patients [28]. Finally, the IFIG expression levels (IFI44, IFI27, RSAD2, and IFI6) were positively correlated with ANA, anti-La, anti-RNP, anti-Ro, and anti-Sm in the whole blood of SLE patients [29]. Thus, the IFN-I signature is usually positively correlated with the levels of autoantibodies in patients with SLE disease.

Complement component 1q (C1q) is a key molecule for complement activation, and low C1q is associated with the development of SLE. The presence of anti-C1q antibodies is found in 20–50% of SLE patients. Purified anti-C1q antibodies from lupus nephritis inhibited phagocytosis of early apoptotic cells, which contributed to the accumulation of autoantigen and the inhibition of clearance of apoptotic cells [30].

Antibodies specific to Ro and La are less frequent in SLE patients than in Sjögren syndrome (SS). Anti-Ro antibodies are associated with cutaneous lupus erythematosus, while anti-La antibodies are related to lupus nephritis and skin disease [25]. In addition, pregnant women with anti-Ro are the most at risk to develop neonatal lupus in the fetus with congenital heart block [31].

## 5. Controversy between Disease Activity and Type I Interferon Signature in SLE

There is some controversy regarding the association between IFNα expression/score and SLE disease activity index (SLEDAI-2K). Positive associations were observed between IFN5 score (LY6E, OAS1, IFIT1, ISG5, and MX1) and renal SLEDAI-2K (>3 criteria) [21], between IFN5 score (EPSTI1, IFI44L, OAS3, and RSAD2) and SLEDAI-2K [14], and between IFN3 score (IFI44, IFIT1, and PRKR) and SLEDAI-2K [4]. In contrast, no correlation was found in SLE patients between SLEDAI-2K and the IFN3 score calculated with IFI27, OAS3, and IFI44 [11]. The IFN-I scores remained stable over 3–12 months despite marked changes in SLEDAI activity [21], suggesting that the IFN-I signature is not synchronous with acute changes in disease activity in SLE.

The contradiction between studies is likely due to the clinical activity of SLE patients (active versus inactive disease) and the diversity of manifestations that changes the SLEDAI-2K scores. In addition, some IFIGs used to measure the IFN-I signature (and IFNα scores) might not be the most responsive to changes in disease activity [32,33]. Depending on the selected IFIGs, the expression of IFNα may not be a dynamic factor of the SLE disease progress but a stable characteristic of the patient and innate activation state of the IFNα pathway.

## 6. American College of Rheumatology (ACR) Criteria and Type I Interferon Signature

The ACR revised a set of criteria for the classification of SLE disease. This classification is based on 11 criteria, including malar rash, discoid rash, photosensitivity, oral ulcer, arthritis, serositis (pleurisy and pericarditis), renal disorder (persistent proteinuria and cellular casts), neurologic disorder (psychosis and seizures), hematologic disorder (hemolytic anemia, leukopenia, lymphopenia, and thrombocytopenia), immunologic disorder, and ANA [34]. Among ACR criteria, proteinuria and renal involvement were not associated with IFN3α scores (measured by PRKR, IFI44, and IFIT1 or by IFIT1, MX1, and PRKR) [21,35]. In contrast, renal involvement in SLE patients was associated with a high IFN3 score (PRKR, IFIT1, and IFI44) [4]. In addition, the IFN4 score (LY6E, OAS1, MX1, and ISG15) was higher in patients with lupus nephritis than in patients without, especially during active renal disease [16]. IFN-I gene expression (MX1, EI2AK2, and IFIT1) in serum from SLE patients was positively correlated with arthritis, nephritis, and lymphadenopathy [36]. Therefore, the selection of IFIGs influences the association between the IFN-I score and ACR criteria.

## 7. Type I Interferon Signature in other Autoimmune Diseases

Elevated IFN-I signature has been reported in other autoimmune diseases, including RA [37], systemic sclerosis (SSc) [38], Sjögren’s syndrome (SS) [39,40], dermatomyositis (DM) [41], and multiple sclerosis (MS) [42].

RA is a systemic autoimmune disease characterized by chronic inflammation in the synovium of the joint tissue and autoantibodies in the serum, especially rheumatoid factor and anti-cyclic citrullinated peptide antibodies [43]. Using microarrays, the IFN43 score (calculated with 43 IFIGs) was higher in 20 RA patients than in 15 RA patients or 15 healthy controls in peripheral blood, suggesting that the IFN-I signature is present in a subgroup of patients with RA, characterized by increased activity of innate immunity, coagulation, and complement cascades [43]. Similarly, a high IFN35 score (calculated with 35 IFIGs using Affymetrix microarray technology) was detected in 22% of RA patients (22/102) in whole blood [37]. This study revealed that the IFN-I signature was heterogeneous in RA patients and the treatment with anti-TNFα reduced IFN-1 expression in RA patients [37]. Finally, IFIG response (IFIT2, RSAD2, STAT1, and XAF1) was overexpressed in fibroblasts and monocytes from RA synovium samples, while T cells had upregulation of interferon regulatory factors (IRFs), including IRF7 and IRF9 [44]. The subgroup of RA patients with an elevated IFN-I signature was more likely to respond to treatment with rituximab and tocilizumab [45,46,47].

Patients with SSc possess endothelial cell dysfunction and immune impairment, resulting in fibrosis and Raynaud’s phenomenon [38]. SSc is also characterized by a dysregulation of humoral immunity and the production of autoantibodies to nuclear and nucleolar components, which can lead to an elevated IFN-I signature. Microarrays showed a higher gene expression of seven IFIGs (G1P3, G1P2, MNDA, IRF7, TAP1, ISG20, and MX1) in peripheral blood cells from patients with SSc than in healthy controls [38]. However, elevated IFIG expression levels in peripheral blood of SSc with established disease (47–68%) were lower than those in SLE [48]. Previous findings suggest that elevated IFN signature could happen later in the SSc disease and play a less critical pathogenic role than in SLE and SS patients [49].

SS (also named sicca syndrome) is a chronic autoimmune disease characterized by lymphocytic infiltration of exocrine glands [50]. SS pathogenesis is associated with autoantibody production and dysregulation of apoptosis. Using microarrays, gene expression of IFIGs was overexpressed in the peripheral blood of SS patients compared to healthy controls [51]. The expression levels for most IFIGs were positively correlated with the titers of anti-Ro/SSA and anti-La/SSB antibodies [51]. Salivary gland biopsy specimens from SS patients contain various immune cells producing IFNα [40]. Sera from SS patients had anti-RBP antibodies (including SSA/Ro, SSB/La, RNP, and/or Sm), which are associated with increasing IFNα production in peripheral blood mononuclear cells. SS and SLE sera had comparable levels of IFNα after stimulation with apoptotic or necrotic cells [40]. Like SLE, SS patients produce immune complexes with IgG binding to nucleic acid released by apoptotic cells, which induce the IFN-I signature in pDCs from peripheral blood [39]. Therefore, when evaluating the specificity of IFN-I signature for SLE diagnosis, SS samples will degenerate more “false positive” during the validation. The specificity could be improved by the selection of IFIGs with higher expression levels in SLE than in SS.

DM is characterized by vascular inflammation, antibodies against endothelial autoantigen, and skin rash [52]. Infiltration of B cells and CD4^+^ T cells was associated with abnormalities in muscle tissue in DM patients. Using microarrays, IFIG expression levels (including ISG5, Mx1, OAS1, IFIT4, IFIT1, IFI44, OAS3, OAS2, IRF7, and IFI27) in muscle samples were higher in DM than in other inflammatory myopathies, such as inclusion body myositis, polymyositis, necrotizing myopathy, and myopathies with inflammation [41]. In addition, the IFN6 score (ISG15, LY6E, IFI44, IFI27, IFIT1, and OAS1) was higher in 13 DM patients than in 40 patients with inflammatory myopathies (no patients had SLE or MCTD). Like SS, DM would reduce the specificity of IFN-I signature in SLE due to its presence in DM patients.

MS is a chronic neurological disorder with heterogeneous demyelination and inflammation in the white matter of the central nervous system. Gene expression of IFIGs (IFI44L, IFITM1, G1P2, IFITM3, and Mx1) in peripheral blood cells was higher in MS than in healthy controls, suggesting an elevated IFN-I signature in MS patients [42]. Moreover, IFNα levels increased in peripheral blood from MS patients compared to healthy controls [53]. The role of the IFN-I signature in MS is still not well understood.

Systemic autoimmune rheumatic diseases (SARDs) (Figure 3), which include SLE, SS, SSc, DM, RA, and PM, have overlapping clinical characteristics, especially the production of ANA. The IFN5 score (EPSTI1, IFI44L, LY6E, OAS3, and RSAD2) was higher in SARD patients with positive ANA than in SARD patients with negative ANA [49]. In early SARD, elevated IFN5 was detected in 65.5% of patients (SLE 80.3%, MCTD/DM 100%, SS 82.6%, and SSc 42.3%). Interestingly, SSc patients had the lowest IFN5 score among the SARD groups. Elevated IFN-I signature was detected in 35% of patients with early SSc, but their IFN5 score was lower than in other early SARD groups. In all ANA-positive groups, only the presence of anti-Ro and anti-La antibodies were positively correlated with the IFN5 score among all autoantibodies (including anti-RNP, anti-Sm, and anti-dsDNA), suggesting that anti-Ro and anti-La play a role in IFNα production [49]. Moreover, a high IFN-I signature was observed in asymptomatic patients that had positive ANA (titer ≥1:160 by immunofluorescence), confirming that elevated levels of IFIG expression are associated with ANA production. Another study reported that the breakdown of positive IFN-I signatures (top five IFIGs were IFI6, RSAD2, STAT2, IFI44, and IFI27) in whole blood of patients were 73% SLE, 68% SSc, 66% DM, 61% polymyositis (PM), and 33% RA [29]. These findings confirm the major role of autoantibodies in the development of IFN-I signature.

As previously discussed, the selection of IFIGs plays an important role in the accuracy of IFN-I signature. Potential criteria of IFIG selection for specific autoimmune diseases are higher expressed IFIGs in the specific disease than other diseases and healthy controls, at least six IFIGs used to calculate the IFN score, and positive correlations between selected IFIGs and autoantibody concentrations. In addition, the gene expression of IFIG(s) that is positively correlated with disease activity/severity is desirable to improve the prognostic of the patient.

IFNs are well known for their antiviral properties and are grouped into two categories, type I (IFNα) and type II (IFNγ) [42]. Viral infection induces the production of IFNα, while activated T and NK cells produce mainly IFNγ. Viral infection in healthy populations can increase IFN-I expression in blood. A high IFN35 score (calculated with 35 IFIGs using microarrays) was detected in 15% of healthy controls (15/100) in whole blood [37]. However, expression levels of “common” IFN-I signature (IFI27, LY6E, and IFI44L) in CD4^+^ T cells and monocytes from PBMCs were higher in SLE than in immunized healthy donors with yellow fever vaccine YFV-17D [54]. Their findings revealed that monocytes (subsets CD16^−^ and CD16^+^) possess a more complex transcriptional regulation in response to IFN-I than CD4^+^ T cells in viral infection [54]. Thus, the IFN-I signatures in SLE and normal immune response against a virus differ in the activation of IFIGs and their expression levels.

## 8. Medical Therapy and Type I Interferon Signature

Specific medical therapy can reduce IFN-I signature in SLE patients. Hydroxychloroquine (HCQ) or corticosteroid therapy showed a trend toward a negative correlation with the IFN-I score (PRKR, IFIT1, and IFI44) in PBMCs of SLE patients [4]. Chloroquine inhibited cell signaling in TLR pathways by reducing complex-mediated cell activation and IFNα production [55]. IFIG expression levels were lower in SLE patients with pulse glucocorticoid (GC) therapy than those without this therapy, likely due to a reduction in IFNα-producing pDC cells [4,56]. The use of immunosuppressive agents was positively correlated with IFN5 (LY6E, OAS1, IFIT1, ISG5, and MX1) in SLE patients [21]. In contrast, the IFN-I score was similar between SLE patients taking immunosuppressive therapies, including mycophenolate and azathioprine, and those without immunosuppressive agents [4]. Prednisone and antimalarial treatments did not influence the IFN5 score in SLE patients [21]. Similarly, no association was detected between treatment with antimalarial or immunosuppressive drugs (azathioprine, mycophenolate, methotrexate, and cyclosporine) and IFN5 score (EPSTI1, IFI44L, LYE6, OAS3, and RSAD2) in SLE patients [14]. The contradiction between studies is likely due to heterogenicity in SLE disease and variability in response to medical therapy among patients. Lastly, the therapy with corticosteroids, HCQ, methotrexate, or other immunosuppressive drugs could reduce the IFN score [13] but more studies are needed to select the most sensitive and specific IFIGs that respond to immunosuppressive treatment in SLE patients.

## 9. Ethnicity and Type I Interferon Signature

In SLE patients, white race was negatively correlated with IFN-I gene expression (MX1, EI2AK2, and IFIT1) [37] and IFN4 (HERC5, IFI27, IFIT1, and RSAD2) [57]. Similarly, the IFN3 score (PRKR, IFIT1, and IFI44) was lower in Caucasians than in other ethnicities (combined African Americans, Asians, and Hispanics) [4]. IFNα concentration was lower in Caucasians than in Africans or Asians and comparable between Caucasians and Hispanics [58]. The role of ethnicity in the development of elevated IFN-I signature remains to be investigated.

## 10. Conclusions

This review article has clinical relevance in autoimmune disease to improve our knowledge of IFN-I signature to monitor and diagnose SLE disease. Elevated IFN-I signature induces DC and B-cell maturation and T-cell activation, leading to autoantibody production. IFNα could represent a promising target for therapeutic intervention in SLE with elevated IFN-I signature. IFIG expression detects cellular activity (IFNα production by immune cells) and the classification of lupus patients that are similar clinically, leading to the development of diagnostic and prognostic biomarkers that will help physicians to select specific biological treatments. Prospective studies are required to assess the role of IFIG and the best combination of IFIGs to monitor SLE disease activity and drug treatments (response to therapy). The influence of genetic and immunological characteristics of individual SLE patients on the IFN-I signature remains to be investigated.

## Figures and Tables

**Figure 1 biomedicines-11-00864-f001:**
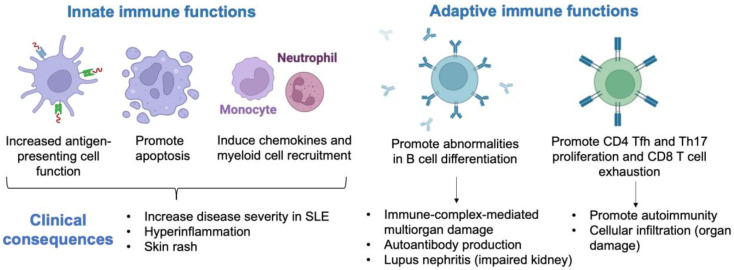
Clinical consequences of type I interferon signature in systemic lupus erythematosus (SLE). Tfh, follicular helper; Th17, T helper 17; CD, cluster of differentiation.

**Figure 2 biomedicines-11-00864-f002:**
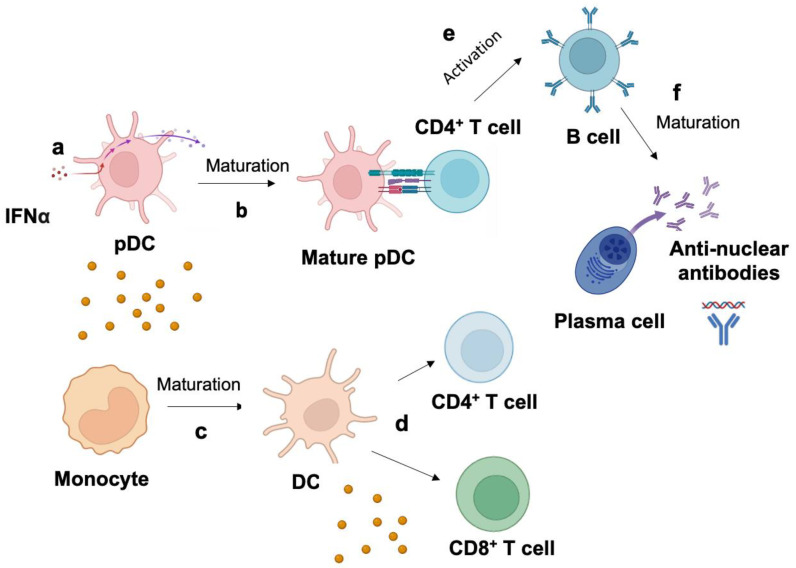
Mechanism of type I interferon signature in systemic lupus erythematosus (SLE) pathogenesis. (**a**) Plasmacytoid dendritic cells (pDCs) produce IFNα due to immune recognition of autoantigens derived from apoptotic material from dead and dying cells and neutrophil extracellular trap debris. (**b**) IFNα promotes the maturation of (**b**) pDC to mature pDC and (**c**) monocyte to dendritic cells (DC), which activates T cells. IFNα promotes (**d**) the exhaustion of CD8^+^ cells and (**e**) the activation of B cells after interacting with a CD4^+^ T cell. (**f**) B cell matures into plasma-cell-producing autoantibodies specific to nucleic-acid-containing autoantigens. High autoantibody production led to the loss of B cell tolerance in SLE.

**Figure 3 biomedicines-11-00864-f003:**
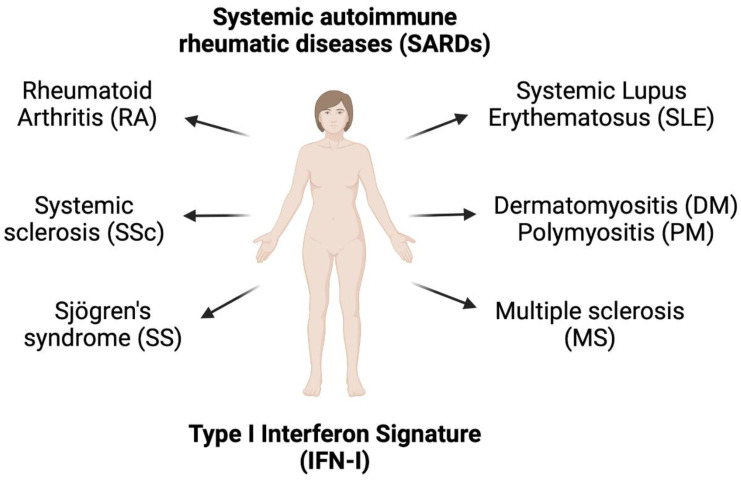
Presence of type I interferon signature in systemic autoimmune rheumatic diseases, which include systemic lupus erythematosus, rheumatoid arthritis, systemic sclerosis, Sjögren’s syndrome, multiple sclerosis, dermatomyositis, and polymyositis.

**Table 1 biomedicines-11-00864-t001:** Common interferon-α-inducible genes in systemic lupus erythematosus patients.

Gene Symbol	Entrez Gene Name	Subcellular Locations ^1^	SLE Subsets
IFI27	Interferon alpha-inducible protein 27	Nucleus, mitochondrion	myeloid cell, T cell
IFI44L	Interferon-induced protein 44-like	Cytosol, nucleus	myeloid cell, T cell, B cell
IFI44	Interferon-induced protein 44	Nucleus, mitochondrion	myeloid cell, T cell, B cell
RSAD2	Radical S-adenosyl methionine domain containing 2	Endoplasmic reticulum, mitochondrion	myeloid cell, T cell, B cell
IFIT1	Interferon-induced protein with tetratricopeptide repeats 3	Cytosol	myeloid cell, T cell
LY6E	Lymphocyte antigen 6 complex, locus E	Extracellular, plasma membrane	T cell
EPSTI1	Epithelial stromal interaction 1	Cytosol	unknown
OAS3	2′-5′-oligoadenylate synthetase 3	Cytosol, nucleus, plasma membrane	myeloid cell, T cell, B cell
OAS1	2′-5′-oligoadenylate synthetase 1	Cytosol, nucleus	T cells
ISG15	ISG15 ubiquitin-like modifier	Cytosol, nucleus, extracellular	myeloid cell, T cell, B cell
PRKR	Platelet-activating factor receptor	Nucleus, extracellular	unknown
SIGLEC1	Sialic acid binding Ig like lectin 1	Extracellular, plasma membrane, endosome	myeloid cells
MX1	Myxovirus (influenza virus) resistance 1	nucleus	myeloid cell, T cell, B cell
HERC5	Hect domain and RLD 5	cytoplasm	myeloid cells

^1^ Subcellular location of each gene was identified using https://www.genecards.org, accessed on 12 January 2023.

**Table 2 biomedicines-11-00864-t002:** Autoantibodies in systemic lupus erythematosus (SLE) and their association with the type I interferon signature (IFN-I), clinical outcomes, and disease activity.

Antibodies ^1^	Frequency in SLE (%)	Association with IFN-I	Clinical Outcomes	Association with Disease Activity
ANA IgG	95	yes	Autoimmune disease	no
Anti-dsDNA IgG	50–60	yes	LN, skin, cerebral	yes
Anti-Smith IgG	20–40	yes	Renal, neurologic, vasculitis disorders	no
Anti-RNP IgG	23–40	yes	Raynaud phenomenon, myositis	yes
Anti-Rib-P IgG	15	yes	LN, autoimmune hepatitis	no
Anti-La/SSB IgG	30–40	yes	LN, skin disease	yes
Anti-Ro/SSA IgG	12–20	yes	Subcutaneous lupus, neonatal lupus	no
Anti-C1q IgG	20–50	no	LN	yes

^1^ ANA, antinuclear antibody; dsDNA, double-stranded DNA; RNP, ribonucleoprotein; Rib-P, Ribosomal P; C1q, complement 1q; LN, lupus nephritis.

## Data Availability

No new data were created in this article.

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
