# Peer review of "Optimal Selection of IFN-α-Inducible Genes to Determine Type I Interferon Signature Improves the Diagnosis of Systemic Lupus Erythematosus"

_biomedicines, 2023, doi:10.3390/biomedicines11030864_

Round 1

Reviewer 1 Report

This manuscript gives a review on applying IFIGs to determine IFN-I signature, which can be used to diagnose the SLE disease and other autoimmune diseases. The author well described IFN-1-relevant immunopathogenesis, clinical consequences, autoantibodies, disease activities, medical therapy, and ethnicity. Overall, this review shows good effort input from the author, and it is a good reference for readers to understand the application of IFIGs. However, some areas should be improved to provide a comprehensive review of this interesting topic. Here are my comments:

1) Line 62. Change the “Type I Interference Signature” to “Type I Interferon Signature”. The “Type I Interferon Signature” is a key term used in this manuscript, and the key term should be consistent throughout the paper to help the reader understand easily.

2) Line 62-94. This section discusses the application of IFIGs in determining IFN-1 signature and introduces RT-qPCR as a common method to measure IFIGs. However, the sensitivity and specificity of IFN-1 signature need optimization before applying them to SLE patients. Indeed, the author suggests that more studies are needed for this optimization (line 92-94). I recommend the author can discuss the application of new technology (apart from RT-qPCR) on revealing novel important IFIGs and indicating potential ways to select IFIGs with better specificity and sensitivity.

3) Line 211. This section shows the IFN-1 signature in other autoimmune diseases. Since the selection of IFIGs plays an important role in the accuracy of IFN-1 signature, the author should discuss potential criteria of IFIGs selection for specific diseases.

In short, this review is very interesting and the writing is well organized for publication.

Author Response

Response to reviewers

I have made all the reviewer-requested changes and hope you agree that the revised manuscript is now suitable for publication. The edited text in the revised manuscript is shown in underlined red using track changes. I have also responded to each of the reviewer comments below. I would like to take this opportunity to thank the reviewers for helping to improve our manuscript.

Reviewer 1

This manuscript gives a review on applying IFIGs to determine IFN-I signature, which can be used to diagnose the SLE disease and other autoimmune diseases. The author well described IFN-1-relevant immunopathogenesis, clinical consequences, autoantibodies, disease activities, medical therapy, and ethnicity. Overall, this review shows good effort input from the author, and it is a good reference for readers to understand the application of IFIGs. However, some areas should be improved to provide a comprehensive review of this interesting topic. Here are my comments:

Line 62. Change the “Type I Interference Signature” to “Type I Interferon Signature”. The “Type I Interferon Signature” is a key term used in this manuscript, and the key term should be consistent throughout the paper to help the reader understand easily.

>> Thank you for finding this mistake. I changed it to “Type I Interferon Signature”. Line 62, p. 2

Line 62-94. This section discusses the application of IFIGs in determining IFN-1 signature and introduces RT-qPCR as a common method to measure IFIGs. However, the sensitivity and specificity of IFN-1 signature need optimization before applying them to SLE patients. Indeed, the author suggests that more studies are needed for this optimization (Lines 92-94). I recommend the author can discuss the application of new technology (apart from RT-qPCR) on revealing novel important IFIGs and indicating potential ways to select IFIGs with better specificity and sensitivity.

>> Thank you for your recommendation. I added a few paragraphs in this section to discuss the different molecular techniques and how to select IFIG to obtain better specificity and sensitivity.

“The most popular molecular methods available for the determination of IFN-I signature in SLE are RT-qPCR, microarrays, and Nanostring technology. Nanostring is useful for screening IFIGs as it can generate expression analysis of hundreds of genes across many samples. Nanostring nCounter is a strongly sensitive technique as it can detect low abundance mRNAs that are not detected by DNA micro-arrays [9], [10]. The sensitivity of the Nanostring nCounter is comparable to RT-qPCR (Taqman assays or SYBR Green) [9]. Results generated by Nanostring showed higher concordance with RT-qPCR than DNA microarray assay [9]. Gene expression analysis using microarrays is less precise than RT-qPCR, but there is a high degree of correlation between the two methods [11,12]. Lastly, Nanostring is a digital barcode technology for the direct multiplexed measurement of gene expression and is less time-consuming but more expensive than RT-qPCR [13].” Lines 79-89, p. 2-3.

“For Nanostring nCounter, the IFN score can be calculated by 2 methods: a z-score-based standardized score and a geomean score. Standardized z-scores for each IFIG are determined using the mean and standard deviation of healthy controls (HC) with the following equation: (z – score for each gene = [(gene count – mean (HC gene expression)] / [standard deviation (HC gene expression)]. For example, the IFN28 standardized score was calculated by summing the 28 z-scores for each sample [10]. Alternatively, the geometric mean or geomean of the counts of each IFIG in each sample is calculated to generate the IFN score. In addition, the IFN5 score (EPSTI1, IFI44L, LY6E, OAS3, and RSAD2) was determined by Nanostring custom array with different ranges of sensitivity (63.6%–83.8%) and specificity (41.7%–67.2%) depending on the outcomes (disease activity state, intake of prednisone, ³ 2 flares, and new immunosuppressive) [14].

For microarrays, the mean and standard deviation for 3 IFIGs (IFI27, OAS3, and IFI44) from 27 HCs were used to calculate the IFN3 score [11]. For each SLE patient, a z-score was calculated for these 3 IFIGs by subtracting the HC mean from the SLE expression value for that IFIG and then dividing the difference by the HC standard deviation. No sensitivity or specificity was reported in this study [11]. This scoring system has also been used in other studies on IFN-I signature using RT-qPCR [15,16].

More studies are needed to understand the importance of selected IFIGs to optimize and improve the sensitivity and specificity (>80%) for the diagnosis of IFN-I signature in SLE patients. Possible ways to improve these parameters are to increase the number of IFIGs (>6 IFIGs used to calculate IFN score) that are highly expressed in SLE patients compared to HC and other autoimmune diseases, such as rheumatoid arthritis RA, systemic sclerosis (SSc), Sjögren’s syndrome (SS), dermatomyositis (DM), and multiple sclerosis (MS).” Lines 109-132, p. 3.

Line 211. This section shows the IFN-1 signature in other autoimmune diseases. Since the selection of IFIGs plays an important role in the accuracy of IFN-1 signature, the author should discuss potential criteria of IFIGs selection for specific diseases.

>> Thank you for this suggestion. I added a paragraph to suggest potential criteria of IFIGs for specific autoimmune diseases.

“As previously discussed, the selection of IFIGs plays an important role in the accuracy of IFN-I signature. Potential criteria of IFIGs selection for specific autoimmune diseases are higher expressed IFIGs in the specific disease than other diseases and healthy controls, at least 6 IFIGs used to calculate the IFN score and positive correlations between selected IFIGs and autoantibody concentrations. In addition, the gene expression of IFIG(s) that is positively correlated with disease activity/severity is desirable to improve the prognostic of the patient.” Lines 442-448, p. 8.

In short, this review is very interesting and the writing is well organized for publication.

>> Thank you so much for your helpful comments.

Reviewer 2 Report

Dear colleague

In the review paper you present the importance of the IFN signature for the differentiation, manifestation and therapy of SLE, which has been shown empirically and was modelled by studies. n addition, overlaps with other autoimmune diseases such as rheumatoid arthritis, systemic sclerosis, Sjögren's syndrome are presented.

Since you work in a commercial institution that provides diagnostic and prognostic testing for patients with SLE, there is an overlap of scientific work with your commercial role.

In the introduction, the second sentence (several SLE manifestations exist) repeats the first and can therefore be deleted.

In Chapter 4, autoantibodies and IFN are discussed. A different order would make sense here. The 2nd paragraph should start with ANA in general (previously paragraph 3) before discussing specific antibodies such as RNP.

In the same paragraph, page 5, line 165 write "low C1q deficiency" as a double negative. One should be removed.

Page 5 line 172: anti-La is not a relevant risk for neonatal lupus, only anti-Ro.

Table 2 anti-La and SSA are mismatched. Correct is anti-La = SSB, anti-Ro = SSA.

Page 6 SLEDAI-2K is an index, the addition of score is a duplication.

Chapter 7, line 218 here is missing the discriminator between the 20 RA patients and the 15 RA patients with different IFN profile.

Line 229 is incorrect. RA patients have arthritis by definition. The study [source 41] describes preclinical patients (without diagnosis of RA), of which those with IFN signature are more likely to develop manifest arthritis and thus definitive diagnosis.

On page 7 starting on line 233, 2 symptoms/manifestations of SSc are explained in parentheses. Other manifestations of other diseases are not explained. Thus, these can be deleted or an explanation of all clinical manifestations should be added in the appendix.

In line 258, the sentence "DM muscle is an inflammatory myopathy characterized by muscle ...inflammation" is poorly worded. M stands for myositis = muscle inflammation, the same is true for inflammatory myopathy. Thus, muscle inflammation is repeated only 3 times.

On page 8 positive ANA in healthy persons is reported. Here the definition of "positive" is missing.  Is this a titer greater than 1:80 or detection at all?

On page 9 starting at line 326, JAK inhibitors are discussed. These have been evaluated for SLE but are not currently being pursued in this indication. There is currently no JAK inhibitor approved for connective tissue diseases. For clarity, I would omit the section.

The sentence line 334 is misleading. Not the IFN group but the therapy reduces the IFN score.

Author Response

Response to reviewers

I have made all the reviewer-requested changes and hope you agree that the revised manuscript is now suitable for publication. The edited text in the revised manuscript is shown in underlined red using track changes. I have also responded to each of the reviewer comments below. I would like to take this opportunity to thank the reviewers for helping to improve our manuscript.

Reviewer 2

Dear colleague

In the review paper you present the importance of the IFN signature for the differentiation, manifestation and therapy of SLE, which has been shown empirically and was modelled by studies. n addition, overlaps with other autoimmune diseases such as rheumatoid arthritis, systemic sclerosis, Sjögren's syndrome are presented.

Since you work in a commercial institution that provides diagnostic and prognostic testing for patients with SLE, there is an overlap of scientific work with your commercial role.

>> Thank you. I added this specification in the conflict of interest. Line 542-544, p. 10.

In the introduction, the second sentence (several SLE manifestations exist) repeats the first and can therefore be deleted.

>> Thank you. I deleted the second sentence in the introduction. Line 30, p. 1.

In Chapter 4, autoantibodies and IFN are discussed. A different order would make sense here. The 2nd paragraph should start with ANA in general (previously paragraph 3) before discussing specific antibodies such as RNP.

>> Thank you for this great suggestion. I added the ANA in general in the 2nd paragraph and the RNP in the 3rd paragraph. Line 188-310, p. 5-6.

In the same paragraph, page 5, line 165 write "low C1q deficiency" as a double negative. One should be removed.

>> Thank you. I removed the word “deficiency”. Line 216, p. 6, p. 5.

Page 5 line 172: anti-La is not a relevant risk for neonatal lupus, only anti-Ro.

>> Thank you. I removed “anti-La” from this sentence (Line 309, p. 6) and in Table 2.

Table 2 anti-La and SSA are mismatched. Correct is anti-La = SSB, anti-Ro = SSA.

>> Thank you. I corrected these mistakes in Table 2.

Page 6 SLEDAI-2K is an index, the addition of score is a duplication.

>> Thank you. I removed “score” after SLEDAI-2K. Line 320, p. 6.

Chapter 7, line 218 here is missing the discriminator between the 20 RA patients and the 15 RA patients with different IFN profile.

>> Thank you. I removed that sentence due to the repetition of the following sentence. Line 359, p. 7.

Line 229 is incorrect. RA patients have arthritis by definition. The study [source 41] describes preclinical patients (without diagnosis of RA), of which those with IFN signature are more likely to develop manifest arthritis and thus definitive diagnosis.

>> Thank you. I removed this portion of the sentence. Line 370-371, p. 7.

On page 7 starting on line 233, 2 symptoms/manifestations of SSc are explained in parentheses. Other manifestations of other diseases are not explained. Thus, these can be deleted or an explanation of all clinical manifestations should be added in the appendix.

>> Thank you. I deleted the 2 symptoms/manifestations of SSc. Line 372-373, p. 7.

In line 258, the sentence "DM muscle is an inflammatory myopathy characterized by muscle ...inflammation" is poorly worded. M stands for myositis = muscle inflammation, the same is true for inflammatory myopathy. Thus, muscle inflammation is repeated only 3 times.

>> Thank you. I corrected this sentence. Lines 409-410, p. 8.

On page 8 positive ANA in healthy persons is reported. Here the definition of "positive" is missing.  Is this a titer greater than 1:80 or detection at all?

>> Thank you for pointing out this mistake. I corrected the sentence for: “a high IFN-I signature was observed in asymptomatic patients that had positive ANA (titer ³1:160 by immunofluorescence)”… Lines 436-437, p. 8.